# Thermal Conductivity Gas Sensor with Enhanced Flow-Rate Independence

**DOI:** 10.3390/s22041308

**Published:** 2022-02-09

**Authors:** Jiayu Wang, Yanxiang Liu, Hong Zhou, Yi Wang, Ming Wu, Gang Huang, Tie Li

**Affiliations:** 1Science and Technology on Microsystem Laboratory, Shanghai Institute of Microsystem and Information Technology, Chinese Academy of Sciences, Shanghai 200050, China; jywang@mail.sim.ac.cn (J.W.); liuyanxiang@mail.sim.ac.cn (Y.L.); zhouhong@mail.sim.ac.cn (H.Z.); wangyi@mail.sim.ac.cn (Y.W.); wm19871110@mail.sim.cn.ac (M.W.); 2University of Chinese Academy of Sciences, Beijing 100049, China; 3China Astronaut Research and Training Center, Beijing 100094, China; huanggang1971@sina.com

**Keywords:** thermal conductivity, flow-rate independent, thermal gas sensor, MEMS sensor, micro heater

## Abstract

In this article, novel thermal gas sensors with newly designed diffusion gas channels are proposed to reduce the flow-rate disturbance. Simulation studies suggest that by lowering the gas flow velocity near the hot film, the maximum normalized temperature changes caused by flow-rate variations in the two new designs (Type-H and Type-U) are decreased to only 1.22% and 0.02%, which is much smaller than in the traditional straight design (Type-I) of 20.16%. Experiment results are in agreement with the simulations that the maximum normalized flow-rate interferences in Type-H and Type-U are only 1.51% and 1.65%, compared to 24.91% in Type-I. As the introduced CO_2_ flow varied from 1 to 20 sccm, the normalized output deviations in Type-H and Type-U are 0.38% and 0.02%, respectively, which are 2 and 3 orders of magnitude lower than in Type-I of 10.20%. In addition, the recovery time is almost the same in all these sensors. These results indicate that the principle of decreasing the flow velocity near the hot film caused by the two novel diffusion designs can enhance the flow-rate independence and improve the accuracy of the thermal conductivity as well as the gas detection.

## 1. Introduction

Currently, as one of the important detection principles of MEMS sensors, thermal detection has been widely used in fluid sensing fields, especially in gas-related detection, such as gas flow [1,2,3], thermal conductivity [4,5,6], shear-stress [7,8,9], and vacuum [10,11,12], etc. Compared to normal chemical sensors, thermal MEMS sensors have the advantages of broad-spectrum, fast response, low power consumption, and long-term stability, becoming a major, long-standing research focus [13,14,15].

In the application of gas flow and thermal conductivity detection, the main problem of thermal MEMS sensors is that the two factors interfere with each other. For thermal flow sensors, the type of gas needs to be known to pre-calibrate the sensor [16,17,18]. If the gas composition varies during usage, output deviations and false alarms will occur, which severely restricts its application. For thermal conductivity detectors (TCD) in gas chromatography (GC) [19,20,21], a typical application of gas thermal conductivity detection, the flow rate variation will increase the noise, decrease the detection limit performance, present inaccurate quantitative test results, and even present false signal peaks. At the same time, miniaturized GC calls for a higher stability of flow rate and temperature [22,23], which requires a better TCD with flow-independence characteristics.

To date, the flow interference on thermal conductivity detection in thermal sensors has received increasing attention and some research has been carried out to solve this problem. These methods can be divided into two main categories: reference and careful design.

The local convective heat transfer intensity affected by the flow rate can be reflected by the thickness of the thermal boundary layer. The schematic diagrams of different methods are illustrated in Figure 1, which displays the thermal boundary layer under static or low velocity (black line), high velocity (blue line), and optimized high velocity (red line). The details are as follows: 

Reference methods include:

1. Deep and shallow groove structure: G. de Graaf et al. demonstrated a design that a gas-conduction-enhanced shallow groove structure compensates with a deep groove to offset the flow impaction effectively [24], as shown in Figure 1a. While this compensation decreases the thermal conductivity sensitivity, the horizontal placement of the structures causes a large dead volume and affects the response speed.

2. Optimized sensing element: By optimizing the position of the temperature sensing element, C.J. Hepp et al. utilized the opposite characteristics of heat convection and heat conduction changing with the flow rate in the constant power mode to stabilize the temperature of the sensing element [25,26], as shown in Figure 1b. This method is simple in structure, but the suitable flow ranges for different gases are varied so that it is limited to applying in gases with similar thermal conductivity.

Careful design methods include:

3. AC excitation and the thermal sublayer: With 200 Hz AC excitation, Reyes Romero et al. proposed a thermal sublayer where the thermal conduct is the dominant heat transfer and the flow rate change can be negligible [27,28], as shown by the red dashed line in Figure 1c. Therefore, the thermal conductivity of the gas is measured by testing the time-of-flight, and its sensitivity needs to be improved.

4. Multiple hot wires: B. C. Kaanta et al. designed a device with multiple hot films arranged in line [29,30,31], as shown in Figure 1d. With the front-stage hot wires preheating the input gas, the latter-stage hot wires can be in the fully developed section, where the thermal boundary layer and the convective heat transfer can remain stable when the input flow rate varies. This design achieves satisfactory results, but the multiple hot wires need to be operated precisely, which is risky in long-term reliability.

5. Barrier or enclosed space: By setting a blocking structure or an enclosed space, the gas flow velocity at the structure location can be greatly reduced and even reach a stagnant state so that the thermal boundary layer and convective heat dissipation can be stabilized [32,33], as shown in Figure 1e. This method is effective but easily leads to complicated packaging or larger dead volume, which affects the response speed.

In this paper, novel designs are proposed to reduce the flow rate interference in TCD simply and effectively. By using different micro diffusion gas channels, the gas flow velocity near the hot film in the designs can be drastically decreased so that the output fluctuation caused by the input flow rate variation can be significantly reduced. Moreover, the novel designs have no impaction on the response speed and the performance of thermal conductivity detection, indicating that they are suitable for rapid gas detection.

## 2. Materials and Methods

### 2.1. Theory

The heat dissipation methods of the hot film include conduction, convection, and radiation.
(1)$$Q_{out}=Q_{cond}+Q_{conv}+Q_{rad}$$
where the heat radiation can be negligible when the working temperature is below 700 K [14]. Heat conduction mainly includes gas heat conduction and solid heat conduction.
(2)$$Q_{gas}=\frac{A}{L}\lambda_{gas}\;\left(T_{w}-T_{g}\right)$$
where *A* is the heat conduction area, *L* is the characteristic length of heat conduction, and *λ_gas_* is the thermal conductivity of the gas. *T_w_* is the temperature of the hot film, and *T_g_* is the temperature of input gas.

The heat loss through convective heat transfer can be expressed as follows [34]:(3)$$Q_{conv}=h_{conv}A\prime \left(T_{w}-T_{g}\right)$$
(4)$$h_{conv}=C\frac{\lambda_{gas}}{L^{\prime }}Re^{1/2}Pr^{1/3}=C^{\prime }\lambda_{gas}U^{1/2}$$
where *A**′* and *L**′* are the characteristic area and length of convective heat transfer, *U* is the gas flow rate, and *Re* is the Reynolds number. *C* is a constant coefficient. The *Pr* is Prandtl Number, which is usually about 0.6–0.7 for gas.

Both the gas thermal conductivity and the gas flow rate can affect the temperature of the hot film and change the device’s output, which results in the flow fluctuation interfering with the gas thermal conductivity detection [21,29,35].

Lowering the gas flow velocity near the hot film can greatly decrease the convective heat transfer change caused by the fluctuation of the input gas flow rate. Therefore, the interference of the input gas flow change can be effectively reduced so that the output signal is only related to the gas thermal conductivity.

### 2.2. Structure Design

Based on the traditional straight structure (Type-I), two kinds of micro thermals conductivity detectors with diffusion gas channels are designed, as shown in Figure 2a. A square gas chamber is set in the middle of the gas channel, corresponding to the hot film structure. There are two types of openings in the chamber, (1) Type-U: the only opening towards the gas outlet, (2) Type-H: the openings on both sides are perpendicular to the gas flow direction. After entering the device, the input gas divides into two parts at the starting position of the gas chamber and merges at the outlet. During this process, the gas flows into the test chamber and reaches the hot film in the form of lateral or reverse diffusion. This method can decrease the gas flow rate near the hot film position significantly and then reduce the interference of the input gas flow rate on the hot film temperature and the output in the detection of thermal conductivity.

The μTCD consists of two substrates: an upper cover with a gas channel groove and a substrate with a suspended hot film structure. The cross-sectional view and the structure diagram of the designed TCD (take Type-I as an example) are shown in Figure 2b,c. The gas channel is formed by bonding the upper cover and substrate, where the hot film structure is suspended by the composite film to detect the change of the thermal conductivity of the input gas. The designed composite film structure is a low-stress multilayer film formed of silicon oxide and silicon nitride, which can withstand higher temperatures. The cross-section of the gas channel is 320 μm in width and 150 μm in height, respectively, which can match the capillary with an inner diameter of 0.01 inch and reduce the dead volume caused by the assembly. The entire gas channel has a minimal volume, which is less than 100 nL for a straight type with a gas channel length of 2000 μm.

### 2.3. Simulation

With the assistance of COMSOL, a multiphysics simulation software, the core part of the design structure has been simulated. Firstly, the gas flow velocity near the center of the hot film is analyzed. Subsequently, the temperature change of the hot film is simulated when the gas flow rate and thermal conductivity change simultaneously. 

The thickness of the silicon upper cover and the silicon substrate is 1000 and 420 μm, respectively. Considering that the lower surface of the device is in direct contact with the fixed heat sink plane and the upper surface is in direct contact with the air, the bottom surface of the substrate is settled to be a constant temperature boundary of room temperature of 20 °C, and the upper surface of the upper cover is provided with a natural convection with heat transfer coefficient of 1 W/(m^2^·K). A constant current density excitation is applied to the hot film in the current-shell module. 

CO_2_ and He are chosen to be the components of the binary gas due to their significant difference in thermal conductivity. The physical properties of the two gases are listed in Table 1. The input gases with different thermal conductivities can be achieved by controlling different ratios of CO_2_ and He. The concentration of CO_2_ in He is set to change from 0% to 100% in a step of 20%. The gas flow rate was changed from 2.5 to 10 sccm in steps of 1.5 sccm.

#### 2.3.1. Flow Velocity

Figure 3a–c shows the flow velocity distribution at 0.5 μm above the hot film in different gas channels with an input He flow of 10 sccm. The maximum flow velocity in traditional Type-I exactly corresponds to the position of the hot film. In contrast, the flow velocity above the hot film in the Type-H and Type-U is significantly decreased due to the diffusion gas channel designs. 

In order to inspect the influence of the structure design on the gas flow velocity near the hot film, with the input He flow ranging from 2.5 to 10 sccm, the gas flow velocity at 0.5 μm above the center of hot film is analyzed, as shown in Figure 3d. The gas flow velocity near the hot film changes approximately linearly with the input gas flow rate in different types. The linear fitting slopes in the Type-U and Type-H are −6.34 × 10^−7^ and 3.47 × 10^−4^ m/s/sccm, respectively, while the Type-I is 0.0130 m/s/sccm. This result indicates that when the input flow rate changes, the flow velocity variation near the hot film is significantly reduced in the novel designed diffusion structures, which provides a basis for the realization of flow-rate independence.

#### 2.3.2. Hot Film’s Temperature

As the electrical resistance of the hot film is proportional to its temperature, the flow interference on the output can be investigated by analyzing the temperature variation of the hot film. The average temperature of the hot film in each type under different gas conditions is simulated, and the results are shown in Figure 4a. 

At the same flow rate, the temperature of the hot film in all types increases with the CO_2_ concentration, indicating that the hot film in the constant current mode can respond to the change of the thermal conductivity of the gas. When the flow rate is 10 sccm, and the CO_2_ concentration is 0%, the traditional Type-I reaches the minimum output, while when the flow rate is 2.5 sccm, and the CO_2_ concentration is 100%, it reaches the maximum value. 

The temperatures in the three types can be normalized to compare the flow rate interference and the normalized temperature output can be calculated by
(5)$$OUT_{Normalized}=\frac{OUT-OUT_{min}}{OUT_{max}-OUT_{min}}\times 100\%$$
where the *OUT* is the temperature of the hot film in the simulation. The results are shown in Figure 4b. Compared with the temperature deviation caused by the flow rate variation in Type-I, there is no obvious change in Type-H and Type-U. Figure 4c plots the normalized temperature deviation caused by the flow rate under different gas conditions, and Table 2 shows the numerical results. The normalized signal deviation caused by the flow-rate increases with the CO_2_ concentration, and its maximum value in Type-I is as high as 20.163%, while it is only 1.220% and 0.017% in Type-H and Type-U, respectively. This result indicates that the diffusion designs can effectively reduce the gas flow rate disturbance to the gas thermal conductivity measurement.

### 2.4. Fabrication

Figure 5a shows the fabrication process of the μTCD, which includes the following steps:A 4-inch n-type single crystal silicon wafer with <100> crystal orientation and 420 μm thickness is used as the substrate material.The SiO_2_/SiN/SiO_2_ low-stress composite film with a thickness of 2 μm is prepared by thermal oxidation and low-pressure chemical vapor deposition (LPCVD) process as the supporting film of the hot wire.Use the lift-off process to prepare 300 nm Pt heating resistor with 200 nm Ti adhesion layer.Use the plasma-enhanced chemical vapor deposition (PECVD) to deposit 400 nm SiN on the surface of the wafer as a passivation layer to improve the stability of the hot wire during high-temperature operation.Use photolithography and reactive ion etching (RIE) to etch windows, and the pipe interface is prepared by TMAH wet etching.Prepare the opening and use the TMAH wet etching process to release the suspended membrane structure.A 4-inch n-type single crystal silicon wafer with <100> crystal orientation and 1000 μm thickness is used as the cover substrate.A 1 μm oxide layer is prepared on the surface by a thermal oxidation process.The alignment mark is etched on the back of the upper cover, which is used for the positive and negative alignment of the subsequent process and the mark required for bonding with the substrate.Use photolithography, RIE, and deep reactive ion etching (DRIE) processes to make the etch window and prepare a deep groove with a depth of 600 μm, which is used as the pipe interface.Use another set of photolithography, RIE, and DRIE processes to prepare the gas channel with a depth of 150 μm.Coat the glass paste on the surface of the upper cover, and perform a heat treatment at 400 °C.The upper cover and lower substrate are aligned and bonded, the bonding temperature is 430 °C, and the bonding time is 30 min.

Figure 5b–d show the optical photographs of the released hot film structure, the upper covers coated with a glass paste and the finished device.

## 3. Results

### 3.1. Characteristics Test

Figure 6a shows the platform used to test the μTCD’s sensitivity of flow rate and thermal conductivity. In the gas control component, the standard gases of He and CO_2_ are accurately controlled by the mass flow controller MFC and connected to the gas pipeline. The dynamic gas distribution method is used to control the total flow rate of the output gas and the proportion of each gas component. The total gas flow rate ranges from 2.5 to 10 sccm in steps of 1.5 sccm, and the proportion of CO_2_ in He is controlled from 0% to 80% in steps of 20%. All standard gases used in the tests were supplied by Shanghai Weichuang standard gas analytical technology, China. The circuit, consisting of the TCD’s hot film and a precision resistor in series, is operated under constant current mode with a precise power source Keysight B2961. The voltage of the hot film is recorded as the output by a data acquisition system, which is composed of Aligent 34972A and a computer.

The binary input gas changes the flow rate and concentration simultaneously, and the test result is shown in Figure 6b. At the same flow rate, the output signals in all types of devices increase approximately linearly with the concentration of the CO_2_ in He. To compare the flow rate interference, the output can be normalized by Equation (5), and the results are shown in Figure 6c. At the same CO_2_ concentration, as the flow rate increases, the output of the traditional Type-I drops significantly, while there are no obvious changes in the output of Type-H and Type-U.

The output changes caused by the flow rate variation under different gas conditions are summarized in Figure 6d and Table 3. The maximum output change accounts for 24.907% of the signal variation range. The deviations in Type-H and Type-U are only 1.510% and 1.650%, respectively.

The above test shows that the impactions of the flow rate on the output signal become more significant with the increase of CO_2_ concentration. Therefore, under the gas condition of pure CO_2_, the flow rate range is further expanded to test and compare the devices.

The flow rate of the input CO_2_ gas is controlled to change from 1 to 20 sccm with steps of 1 sccm continuously. Four devices of each type are tested. Figure 7 shows the output change results, which are normalized with the 1 sccm condition as the benchmark. The insert enlarges the details of Type-H and Type-U. It can be seen that as the flow rate increases, the output signal of the traditional Type-I gradually deviates. For Type-H, the output changes less at low flow rates but gradually increases when the flow rate exceeds 10 sccm. While for the Type-U, the output remains stable throughout the flow rate range. The average normalized output deviation caused by flow rate for Type-I is up to 10.201%, while it is only 0.384% and 0.0215% for Type-H and type-U, respectively, exhibiting an excellent characteristic of flow-rate independence. 

These results indicate that compared with the traditional straight type (Type-I), the two diffusion designs can reduce the flow rate interference significantly. Moreover, the two diffusion designs have good consistency in terms of the flow-independence.

### 3.2. Response Speed Test

Since the diffusion gas channel designs reduce the flow velocity in the gas chamber, which may affect the response speed of the device. Therefore, the response speed of different types is tested. When the detector runs at a steady state under the condition of a He flow rate of 2.5 sccm, a few milliliters of CO_2_ gas is quickly injected into the pipeline by a syringe, and the output signal is quickly collected by the NI high-speed data acquisition instrument.

The recovery time, the time required for the output to change from 90% to 10% of the peak value, is used to precisely compare the response speed of different types. Six devices are tested for each type of structure, and the results are shown in Figure 8. The average recovery time in Type-I, Type-H, and Type-U are 168.72, 170.92 and 172.42 ms, respectively. Compared with the straight type, the recovery time in the two diffusion designs change less than 2.2%, which benefits from the extremely small dead volume in the designs. This result indicates that the diffusion-designed μTCD still has the advantage of the fast response speed and is suitable for fast gas detection.

## 4. Conclusions

In summary, novel diffusion designs are proposed to reduce flow-rate interference in thermal conductivity gas sensors simply and effectively. Simulations suggested that the diffusion designs can effectively decrease the gas flow velocity near the hot film. The maximum normalized temperature changes caused by the input flow in Type-H and Type-U are only 1.22% and 0.02%, respectively, while it is 20.16% in the traditional Type-I. The tests show that under different thermal conductivity gas conditions, the maximum normalized output deviation is reduced to 1.51% and 1.65% in Type-H and Type-U, compared to the 24.91% in Type-I. Furthermore, under the input 1–20 sccm CO_2_ flow, the average output deviations caused by flow rate are 0.38% and 0.02% in Type-H and Type-U, which is 10.20% for Type-I. In addition, the diffusion designs have basically no impaction on the recovery time and maintain the advantage of the fast response speed. The principle of decreasing the flow velocity near the hot film introduced by the two novel diffusion designs can enhance the flow-rate independence and improve the accuracy of gas thermal conductivity detection. 

## Figures and Tables

**Figure 1 sensors-22-01308-f001:**
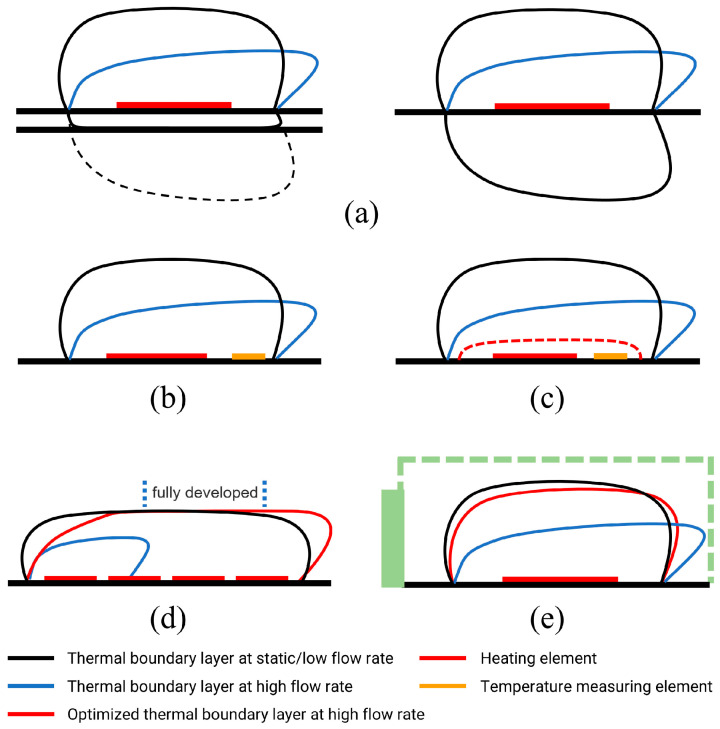
Schematic of the methods to reduce the flow rate interference. (**a**) Deep and shallow groove structure, (**b**) optimized sensing element, (**c**) AC excitation and the thermal sublayer, (**d**) multiple hot wires, (**e**) barrier or enclosed space.

**Figure 2 sensors-22-01308-f002:**
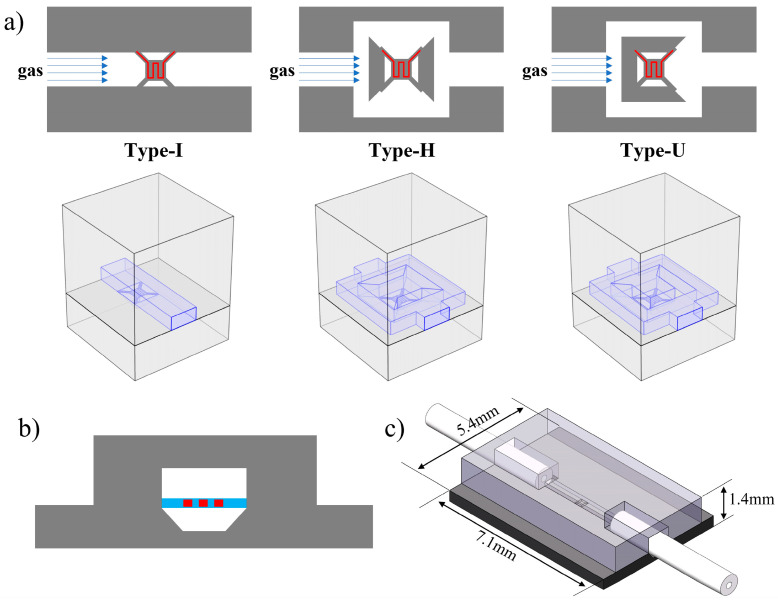
The schematic diagram of the designed structures: (**a**) Top view of different gas channel designs (Type-I, Type-H and Type-U) and the 3D models for simulation, (**b**) cross-section view of the TCD, (**c**) structure diagram of the designed TCD (take Type-I as an example).

**Figure 3 sensors-22-01308-f003:**
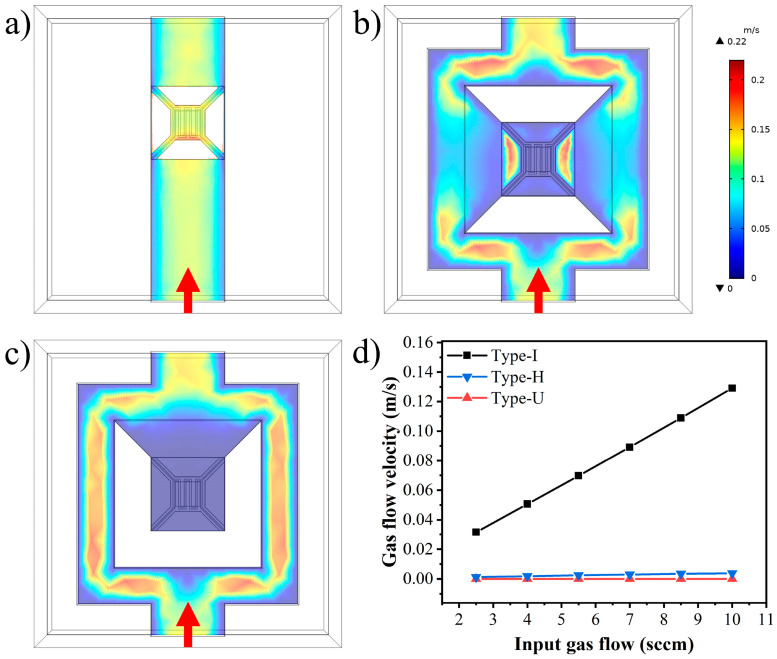
The simulation results: the flow velocity distribution at 0.5 μm above the hot film in (**a**) Type-I, (**b**) Type-H, (**c**) Type-U, the arrows indicate the flow direction, (**d**) the gas flow velocity at 0.5 μm above the center of the hot film in different designs.

**Figure 4 sensors-22-01308-f004:**
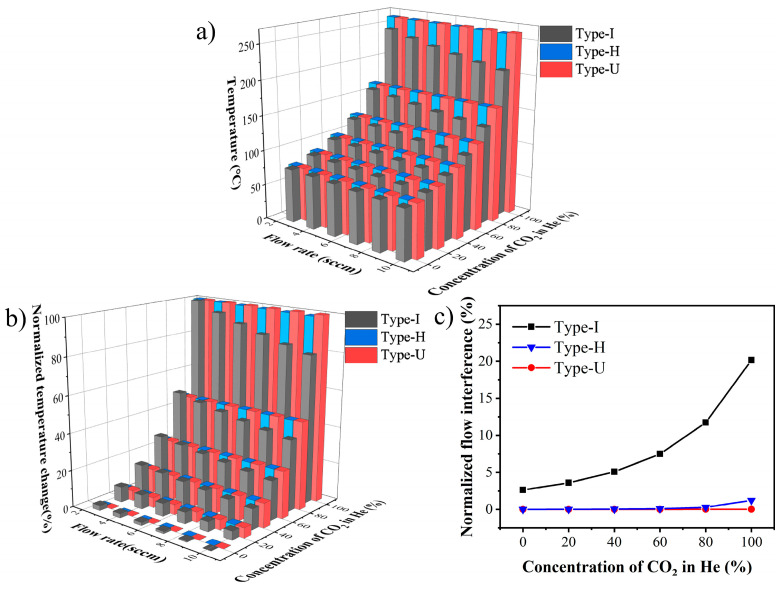
The simulation results of changes in gas flow rate and concentration simultaneously. (**a**) The temperature of the hot film, (**b**) the normalized temperature, (**c**) the normalized flow interference under different gas conditions.

**Figure 5 sensors-22-01308-f005:**
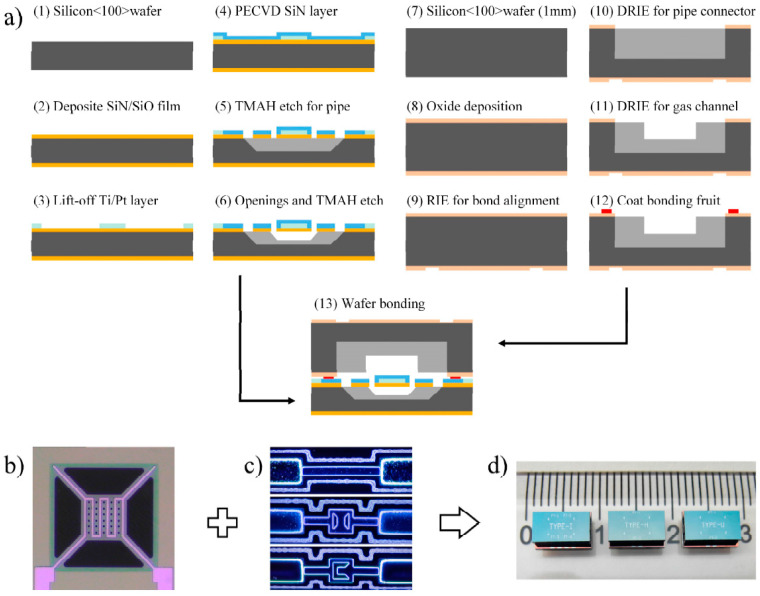
Fabrication process and optical photographs of the fabricated device. (**a**) the fabrication process, (**b**) the released hot film structure, (**c**) the covers coated with glass paste: Type-I (upper), Type-H (middle) and Type-U (lower), and (**d**) the finished device (unit: cm).

**Figure 6 sensors-22-01308-f006:**
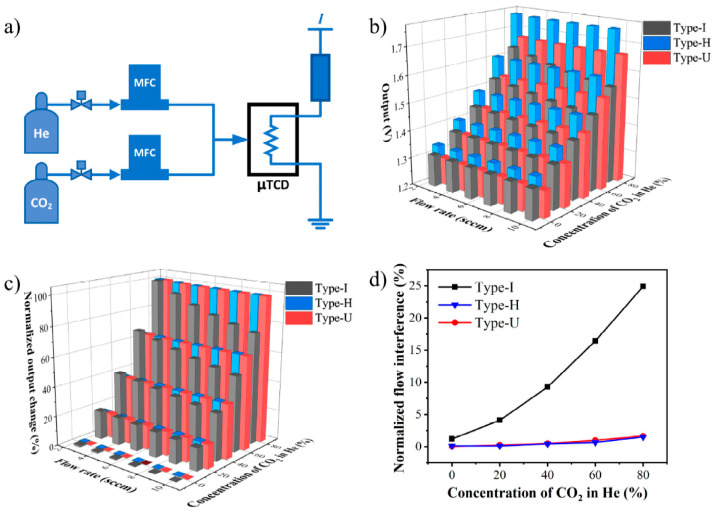
The experiment of changes in gas flow rate and concentration simultaneously. (**a**) The test platform, (**b**) the output of the devices, (**c**) the normalized output change, (**d**) the normalized flow interference under different gas conditions.

**Figure 7 sensors-22-01308-f007:**
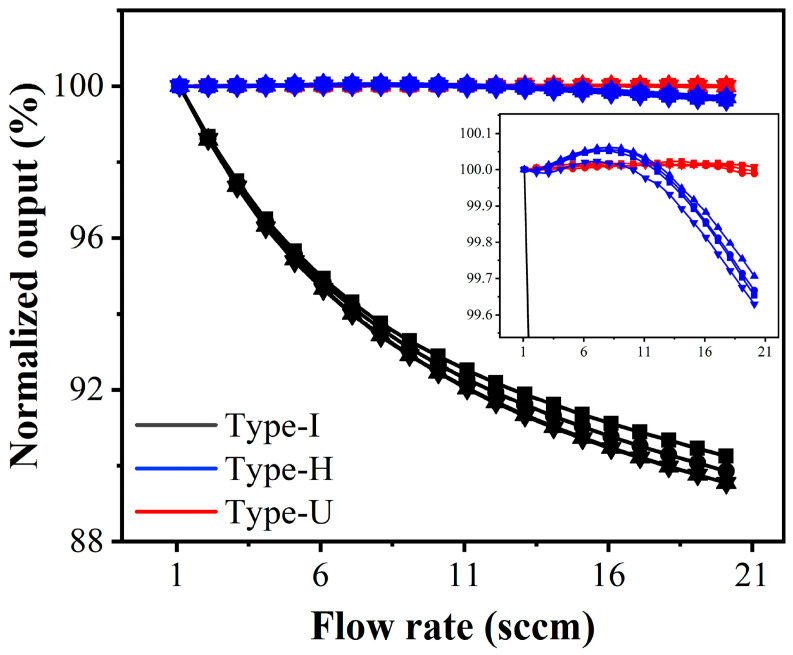
The normalized output change in different types of devices with the CO_2_ flow ranging from 1 to 20 sccm.

**Figure 8 sensors-22-01308-f008:**
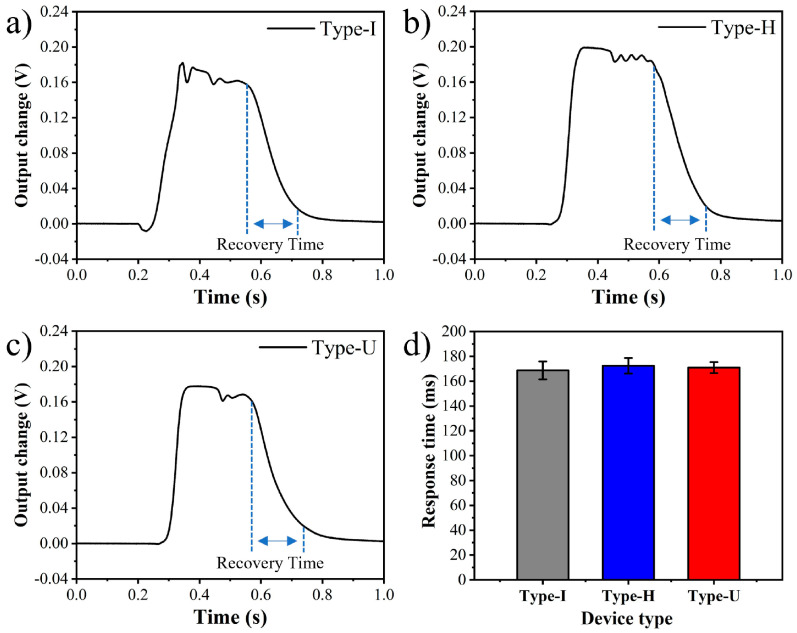
The response curve of (**a**) Type-I, (**b**) Type-H, (**c**) Type-U, (**d**) the recovery time of different types.

**Table 1 sensors-22-01308-t001:** Thermal properties of He and CO_2_ at *p* = 1 bar [36].

Gas Type	Temperature (K)	Thermal Conductivity (mW/(m·K))	Density (kg/m^3^)	Specific Heat Capacity (kJ/(kg·K))
He	300	156.0	0.1604	5.193
	400	190.4	0.1203	5.193
	500	222.3	0.09626	5.193
CO_2_	300	16.79	1.773	0.8525
	400	25.14	1.326	0.9417
	500	33.49	1.059	1.015

**Table 2 sensors-22-01308-t002:** Simulation results: the normalized temperature change of the hot film under multiple concentrations.

CO_2_ in He	Type-I	Type-H	Type-U
0%	2.626%	0.005%	0.003%
20%	3.593%	0.014%	0.004%
40%	5.074%	0.035%	0.004%
60%	7.483%	0.091%	0.006%
80%	11.727%	0.282%	0.009%
100%	20.163%	1.220%	0.017%

**Table 3 sensors-22-01308-t003:** Test results: the normalized flow-rate interference on the output signals.

CO_2_ in He	Type-I	Type-H	Type-U
0%	1.195%	0.116%	0.032%
20%	4.101%	0.116%	0.259%
40%	9.277%	0.434%	0.488%
60%	16.426%	0.669%	0.986%
80%	24.907%	1.510%	1.650%

## Data Availability

Not applicable.

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
