# Peer review of "Thermal Conductivity Gas Sensor with Enhanced Flow-Rate Independence"

_sensors, 2022, doi:10.3390/s22041308_

Round 1

Reviewer 1 Report

This manuscript reported a gas sensor based on enhanced flow rate independence. The topic is interesting. However, it needs further improvement before publishing.

1, Any specific reason for flow rate interference in Figure 1? What is the difference between them? any reference to support the difference? or any theory on this six figures?

2, The three types were designed by authors in Figure 2. The authors can show real figures of the design in Figure 2

3, What is the response in Figure 6? Can the author show the dynamic measurement data in Figure 6?

4, What is the reason for the gas response?

5, How does the humidity and temperature to the device in gas response?

Reviewer 2 Report

Please find my comments attached

Round 2

Reviewer 1 Report

The authors did answer my questions well. I recomment to publish in the current form.